# SITU: A Simple Training-Free Thinking-with-Image Approach via Uncertainty Guidance

## Abstract

Large Multimodal Models (LMMs) have shown great promise in complex reasoning by incorporating images as intermediate steps, a paradigm known as "thinking with images". However, most current "thinking with images" techniques are training-based, incurring significant computational costs, limiting model versatility, and risking catastrophic forgetting. To bridge this gap, we propose `SiTu`, a simple, training-free framework for "thinking with images" that leverages an LMM's inherent uncertainty to achieve test-time scaling for multimodal reasoning. The core of our approach is the discovery of a stable, entropy-based uncertainty estimation native to LMMs, which reliably guides the dynamic combination of diverse perception enhancement paths. We implement three simple perceptual actions, categorized as visual highlighting and irrelevant information suppression, and demonstrate a notable scaling phenomenon: as the number and diversity of these actions increase, the LMM's reasoning ability improves consistently. Our extensive experiments on fine-grained visual understanding benchmarks like $V^*$, HR-Bench 4K, HR-Bench 8K, and MME-realworld show that `SiTu` significantly outperforms existing training-free perception enhancement methods. Surprisingly, `SiTu` even surpasses the performance of current state-of-the-art training-based "thinking with images" methods, highlighting the immense potential of test-time scaling for multimodal reasoning.

## 1 Introduction

Large Multimodal Models (LMMs) have advanced rapidly, enabling complex multimodal reasoning. A promising paradigm, "thinking with images" (Su et al., 2025b; Zheng et al., 2025b; Zhang et al., 2025c; Zhu et al., 2025; Su et al., 2025a; Huang et al., 2025; Liu et al., 2025; Zhang et al., 2025b; Zhou et al., 2025; Zhang et al., 2025a; Wang et al., 2025a; Bai et al., 2025b; Ni et al., 2025), has recently emerged, where LMMs generate and incorporate images as intermediate steps to enhance multimodal reasoning. This novel approach empowers models to iteratively refine their visual understanding, ultimately yielding more precise and reliable results. However, most current "thinking with images" techniques are training-based, relying on specialized datasets and intensive fine-tuning. This pipeline, while effective, presents several significant limitations. First, it imposes a considerable computational cost; for example, the high-performing method by Zhang et al. (2025d) required 1,200 GPU hours on 32 NVIDIA H800 GPUs. Second, the fine-tuning process can narrowly specialize a model's capabilities, limiting its versatility. An example is the approach by Ni et al. (2025), where fine-tuning on a point-prompting dataset restricted the model to this singular reasoning strategy. Lastly, fine-tuning risks catastrophic forgetting, may leading to the degradation of the model's fundamental perception and reasoning abilities developed during massive pre-training.

In contrast, the field of Large Language Models (LLMs) has seen a different promising paradigm: training-free test-time scaling (Muennighoff et al., 2025; Fu et al., 2025). Methods such as Chain-of-Thought (CoT) (Wei et al., 2022), Best-of-N sampling (Freitag & Al-Onaizan, 2017), and self-consistency (Wang et al.) dramatically improve reasoning accuracy without requiring additional training or external tools by just generating extra tokens at inference time. Their low-cost and high-transferability make them a compelling choice for a wide range of downstream tasks.

However, we find that these highly effective training-free test-time scaling methods unexpectedly fail when applied to LMMs, particularly on fine-grained visual understanding tasks (Wu & Xie, 2024;

Wang et al., 2024) that demand focused attention on specific visual subregions to answer detailed questions. As depicted in Figure 1, these methods only show slow, small-scale improvements, and Zero-shot CoT even exhibits performance degradation as it scales. This uniform underperformance reminds us to consider the fundamental differences between LMMs and LLMs and their respective tasks. Prior work has highlighted that, unlike text, which is a discrete input space, visual information exists in a continuous space, making the task of grounding continuous visual signals into discrete semantic tokens significantly more challenging for LMMs (Yang et al., 2024). Furthermore, inaccurate perception in multimodal tasks can lead to catastrophic and often unrecoverable errors in subsequent reasoning (Su et al., 2025b). We argue that the limitations of existing training-free methods in the multimodal domain arise from their failure to improve the fundamental perceptual abilities of LMMs and, at times, from their propensity to impair them. Another line of training-free methods attempts to enhance LMM perception through agent-like workflows (Li et al., 2025; Shen et al., 2024; Wang et al., 2025b). However, these approaches are with two main limitations: First, their fixed workflows require meticulous, task-specific design and prompt engineering to be effective across different LMMs. Second, their predetermined steps can be restrictive, sometimes even hindering a model's capabilities on unsuitable tasks and leading to performance degradation. Motivated by these observations, we pose the following question: *Can we find a common training-free test-time scaling method for the multimodal domain through enhancing LMM perceptual ability?*

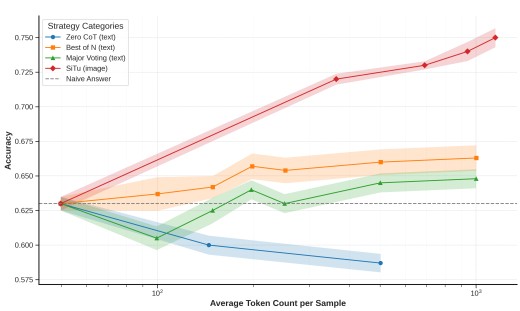

Figure 1: Comparison of existing training-free test-time scaling methods on fine-grained visual understanding benchmark HR-Bench 8K. The x-axis represents the number of tokens generated, and the y-axis shows the accuracy change. "Naive Answer" represents a direct, unassisted LMM response.

To address this question, we propose `SiTu`, a simple training-free thinking-with-image framework that leverages an LMM's inherent uncertainty to achieve test-time scaling for multimodal domain, especially for challenging fine-grained visual understanding tasks. The cornerstone of our approach is the discovery of a stable, entropy-based uncertainty estimation method that is native to LMMs. This method does not require multiple forward passes or a separate prediction set, yet it can reliably evaluate the confidence of an LMM's response to a given input. Furthermore, we found this uncertainty metric is not only effective for assessing the final answer but also for guiding optimal perceptual paths during the reasoning process. This finding makes it possible to integrate a wide variety of perceptual enhancement operations into a single, unified framework. We also observe a notable scaling phenomenon within our framework: as the number and diversity of these perceptual operations increase, the overall multimodal reasoning ability of the LMM shows a consistent and stable improvement.

To validate our framework, we defined and implemented three simple perception actions categorized into visual highlighting and irrelevant information suppression. For Visual Highlighting, we utilize the LMM's grounding ability to pinpoint and emphasize key objects relevant to the question. For Irrelevant Information Suppression, we employ three distinct strategies to remove irrelevant visual regions, thereby focusing the model's attention on the most critical information. Our experiments on three fine-grained visual understanding datasets demonstrate that our simple `SiTu` method outperforms not only existing LMM perception enhancement methods but also all current open-source training-based "thinking with images" approaches. This remarkable result underscores the significant potential of training-free test-time scaling for LMMs, proving that enhanced performance can be unlocked directly from the models themselves without costly fine-tuning.

Our primary contributions are summarized as follows: 1) We propose `SiTu`, a simple but effective training-free paradigm for "thinking with images" that achieves significant performance gains without costly fine-tuning or architectural modifications. 2) We discover and validate a universal, LMM-native uncertainty metric, which serves as a robust guidance signal for dynamically selecting optimal perceptual enhancement paths. 3) Experiments demonstrate `SiTu`'s state-of-the-art performance on fine-grained visual understanding benchmarks, where it outperforms all existing training-based "thinking with images" and training-free perception enhancement methods.

**Simple Thinking-with–image via Uncertainty Guidance**

Figure 2: **Overview of SiTu.** Our approach explores multiple perceptual actions through Perception Action Space and Multi-strategy Exploration. Then through Uncertainty-guided Selection, identifies and returns the optimal answer with lowest uncertainty metric (the starred answer in the figure).

## 2 METHODS

### 2.1 PARADIGM FORMULATION

Thinking with images paradigm models multimodal reasoning as a sequential process where an LMM dynamically generates interleaved visual and textual intermediate outputs. At each reasoning step $t$, the evolving state history is captured by the sequence of previous multimodal outputs $S_t = (z_1, \ldots, z_{t-1})$. The next reasoning step, $z_t$, is generated by the LMM, conditioned on this history, the initial input image $I$, and the user query $Q$. Formally, we define the set of all possible intermediate states as the union of textual outputs $\mathcal{T}_{\text{text}}$ and visual artifacts $\mathcal{I}_{\text{vis}}$. The model then samples the next state $z_t$ from its conditional probability distribution:

$$z_t \sim P(\cdot|S_t, I, Q; \Theta_{\text{LMM}}), \quad \text{where } z_t \in \mathcal{T}_{\text{text}} \cup \mathcal{I}_{\text{vis}}. \tag{1}$$

### 2.2 UNCERTAINTY-GUIDED SELECTION

The most critical component of our framework is the Uncertainty-Guided Selector, which evaluates the quality of reasoning paths and plays a crucial role in eliminating incorrect answers. To handle open-ended questions and compare confidence across different strategies, we quantify the uncertainty of a generated answer using token-based Shannon Entropy.

The uncertainty $\mathcal{U}(A)$ for an answer candidate $A$ is the average entropy of its tokens. This is calculated as:

$$\mathcal{U}(A) = \frac{1}{N} \sum_{i=1}^{N} \mathcal{H}(t_i) = -\frac{1}{N} \sum_{i=1}^{N} \sum_{j=1}^{V} p_{i,j} \log p_{i,j}, \tag{2}$$

where $t_i$ is the $i$-th token, $N$ is the total number of tokens in the answer, and $p_{i,j}$ is the probability of the $j$-th token in the vocabulary for the $i$-th token position. A lower uncertainty score indicates a more confident and potentially more accurate answer. The final answer $A_{\text{final}}$ is selected by choosing the strategy with the minimum uncertainty:

$$A_{\text{final}} = \arg\min_{A \in \mathcal{A}} \mathcal{U}(A), \tag{3}$$

Crucially, the token probabilities $p_{i,j}$ are derived from the same conditional probability distribution $P(\cdot|S_t, I, Q; \Theta_{\text{LMM}})$ defined in Equation (1). This design ensures that our uncertainty metric is context-aware, incorporating the full multimodal history and input image. Thus, this method provides a promising way to evaluate the effectiveness of our perception-enhancing operations.

**Theoretical Justification: Entropy vs. Confidence.** While related to sequence confidence, Entropy offers a stricter optimization objective for multimodal ambiguity. Mathematically, the Shannon

Table 1: Comparison of our SiTu with existing works on fine-grained visual understanding benchmarks. Open-source models with the best performance in each task are shown in **bold**, the second-best performance is underlined.

| Method | $V^*$ Bench | | | HR-Bench 4K | | | HR-Bench 8K | | |
|---|---|---|---|---|---|---|---|---|---|
| | Attribute | Spatial | Overall | FSP | FCP | Overall | FSP | FCP | Overall |
| Closed-source MLLMs | | | | | | | | | |
| O3 (Hurst et al., 2024) | - | - | 95.7 | - | - | - | - | - | - |
| GPT 4o (Hurst et al., 2024) | - | - | 66.0 | 70.0 | 48.0 | 59.0 | 62.0 | 49.0 | 55.5 |
| QWen-VL-max (Bai et al., 2023b) | - | - | - | 65.0 | 52.0 | 58.5 | 54.0 | 51.0 | 52.5 |
| Open-source MLLMs | | | | | | | | | |
| LLaVA-v1.6-7B (Liu et al., 2024b) | 60.9 | 63.2 | 61.8 | 49.0 | 46.8 | 47.9 | 37.3 | 44.3 | 40.8 |
| LLaVA-v1.6-13B (Liu et al., 2024b) | 60.0 | 64.5 | 61.8 | 49.8 | 41.3 | 45.5 | 38.0 | 38.3 | 38.1 |
| LLaVA-HR-X-7B (Luo et al., 2024) | 51.3 | 64.5 | 56.5 | 57.8 | 46.3 | 52.0 | 42.0 | 41.3 | 41.6 |
| InternVl-1.5-26B (Chen et al., 2024b) | - | - | - | 69.5 | 51.8 | 60.6 | 69.3 | 48.5 | 57.9 |
| Yi-VL-34B (Young et al., 2024) | - | - | - | 46.0 | 42.8 | 44.4 | 39.5 | 38.5 | 39.0 |
| Qwen2.5-VL-7B (Bai et al., 2025a) | 73.9 | 67.1 | 71.2 | 85.2 | 52.2 | 68.8 | 78.8 | 51.8 | 65.3 |
| Training-based Thinking with Image | | | | | | | | | |
| SEAL (Wu & Xie, 2024) | 74.8 | 76.3 | 75.4 | - | - | - | - | - | - |
| Pixel Reasoner (Su et al., 2025a) | - | - | 84.3 | - | - | - | - | - | - |
| Chain-of-Focus (Zhang et al., 2025c) | - | - | 88.0 | - | - | - | - | - | - |
| Simple O3 (Wang et al., 2025c) | - | - | 90.4 | - | - | 76.2 | - | - | - |
| DeepEyes (Fu et al., 2025) | 91.3 | 88.2 | 90.1 | 91.3 | 59.0 | 75.1 | 86.8 | **58.5** | 72.6 |
| Thyme (Zhang et al., 2025d) | 83.5 | 80.3 | 82.2 | 91.0 | 63.0 | 77.0 | 86.5 | 57.5 | 72.0 |
| Training-free methods | | | | | | | | | |
| DyFo (Li et al., 2025) | 80.0 | 82.9 | 81.2 | - | - | - | - | - | - |
| RAP (Wang et al., 2025b) | 80.0 | 84.2 | 81.7 | 80.3 | 42.3 | 61.3 | 81.8 | 45.3 | 63.5 |
| ZoomEye (Shen et al., 2024) | 93.9 | 85.5 | 90.6 | 84.3 | 55.0 | 69.6 | 88.5 | 50.0 | 69.3 |
| SiTu (Our methods) | **94.8** | **88.3** | **92.1** | **95.0** | **64.0** | **79.5** | **92.0** | 58.0 | **75.0** |
| Δ (vs Qwen2.5-VL-7B) | +20.9 | +20.9 | +20.9 | +9.8 | +11.8 | 10.7 | +13.2 | +6.2 | +9.7 |

Entropy $H(p)$ decomposes into two terms:

$$H(p) = \underbrace{-p(\hat{y}) \log p(\hat{y})}_{\text{Inverse Confidence}} + \underbrace{\sum_{y' \neq \hat{y}} -p(y') \log p(y')}_{\text{Distraction / Ambiguity}} \qquad (4)$$

Standard metrics like MaxProb only optimize the first term. However, in fine-grained perception, LMMs often suffer from *Inter-Class Conflict* (e.g., assigning 51% probability to "sedan" and 49% to "coupe"). Entropy explicitly captures this "distraction" (the second term), serving as a more sensitive proxy for perceptual hallucinations.

## 2.3 MULTI-STRATEGY EXPLORATION

To validate the importance of enhanced perception for test-time scaling method in multimodal tasks, our framework explores a set of diverse perception operation. Each reasoning path consists of a sequence of perception operations followed by a final reasoning step. The core idea is that different paths, by enhancing perception in distinct ways, will provide the LMM with a wider range of possible candidates. We formally define a perception operation as a function $\pi \in \Pi$ that takes the current multimodal context—the image $I$ and the query $Q$—and generates an enriched context for subsequent reasoning.

$$(I', D) = f_\pi(I, Q), \qquad (5)$$

where $I'$ is a new visual artifact (e.g., a cropped image or a highlighted region) and $D$ is a textual description. A key advantage of our approach is that these operations do not rely on external expert models or specialized tools; instead, they leverage the intrinsic visual understanding of the LMM itself to create the new context.

A complete reasoning path is a sequence of these operations, culminating in a final reasoning step $\rho$ that generates the answer. We represent a path as:

$$\text{Path}_i = \langle \pi_1, \pi_2, \ldots, \pi_n, \rho \rangle, \qquad (6)$$

The simplest path is the naive one, which consists only of the final reasoning step $\rho$, corresponding to a direct answer from the LMM without any perceptual manipulation. More complex paths involve one

Table 2: Comparison of the `SiTu` against the baseline LMM on the MME-RealWorld benchmark. MO: Monitoring; AD: Autonomous Driving. The "$\Delta(\uparrow)$" represents the performance gains of our RAP against the baselines.

| Method | MO | | | | AD | | | | OCR | | | |
|---|---|---|---|---|---|---|---|---|---|---|---|---|
| | Calculate | Intention | Color | Location | Attention | Attribute | Visual | Relation | Advertise. | License | Address | Text |
| Qwen-2.5-VL-7B | 20.7 | 13.3 | 32.9 | 29.0 | 37.8 | 19.7 | 60.2 | 28.3 | 87.3 | 88.5 | 87.7 | 85.2 |
| -w/ `SiTu` | **30.3** | **25.5** | **51.7** | **36.0** | **48.8** | **21.9** | **64.2** | **30.2** | **88.6** | **90.6** | **89.1** | **85.7** |
| $\Delta(\uparrow)$ | +10.4 | +12.2 | +18.8 | +7.0 | +11.0 | +2.2 | +4.0 | +1.9 | +5.4 | +2.1 | +1.4 | +0.5 |

or more perception operations $\pi_j$ to guide the LMM toward a better understanding before generating the final answer. Each path produces a candidate answer $A \in \mathcal{A}$, and the best answer $A_{\text{final}}$ is then selected from the set $\mathcal{A}$ using the uncertainty metric defined in the previous section.

## 2.4 Perception Action Space

**Draw Boxes Strategy.** This method uses the LMM to perform zero-shot object detection to implement visual highlighting. It prompts the LMM to identify a set of objects $O = \{o_1, o_2, \dots\}$ from the query $Q$ and return their corresponding bounding box coordinates in a structured format. These coordinates are then used to render visual annotations directly on the image. We define a function $\Phi_{\mathbf{b}} : \mathcal{I} \times \mathcal{Q} \to \mathcal{P}(\mathbf{b})$ that queries the LMM to extract a set of bounding boxes $\mathbf{B}$ from an image $I$ based on a query $Q$. Here, $\mathcal{I}$ is the set of all images, $\mathcal{Q}$ is the set of all queries, and $\mathbf{b}$ is the set of all bounding boxes. $\mathcal{D}$ is a rendering function that overlays the boxes in $\mathbf{B}$ onto the image $I$. The operation is then given by:

$$\mathbf{B} = \Phi_{\mathbf{b}}(I, Q) \tag{7}$$

$$I' = \mathcal{D}(I, \mathbf{B}) \tag{8}$$

**Grounding Crop Strategy.** This strategy effectively focuses the model's attention by cropping the image to a region of interest, with irrelevant information suppression as motivation. It prompts the LMM for bounding boxes of objects $O$ mentioned in the query. It then computes a single, unified bounding box $\mathbf{b}_{\text{union}}$ that encompasses all found objects. The final image $I'$ is a cropped version of the original, centered on this unified region. The operation is defined as:

$$\mathbf{b}_{\text{union}} = \cup_{\mathbf{b} \in \Phi_{\mathbf{b}}(I, Q)} \mathbf{b} \tag{9}$$

$$I' = \mathcal{C}(I, \mathbf{b}_{\text{union}}) \tag{10}$$

where the union operation $\cup$ is performed over the bounding box coordinates, and $\mathcal{C}$ is a function that performs the final image transformation.

**Grid Crop Strategy.** This method employs a simple search to locate and crop key objects by partitioning the image, also categorized as irrelevant information suppression. It bypasses traditional object detection by asking the LMM to judge the presence of a target object $o$ within different image segments. Specifically, the image is partitioned into an even grid of segments. We define a function $\Phi_P : \mathcal{I} \times \mathcal{O} \to [0, 1]$ that queries the LMM for the probability $P(o \in S)$ of an object $o$ being in an image segment $S$. The process for a single object is to find the most probable segment $S^*$:

$$S^* = \arg\max_{S \in \mathcal{S}} \Phi_P(S, o) \tag{11}$$

where $\mathcal{S}$ is an even grid partition of the image $I$. The final image $I'$ is a composite of the most probable regions found for all objects, effectively solving a localization problem through a simple, coarse-to-fine search process.

## 3 Experiments

### 3.1 Setups

**Benchmarks and Metrics.** To thoroughly evaluate our framework, we conduct experiments on fine-grained visual understanding and high-resolution multimodal datasets. The V$*$ Bench and

Table 3: Cross-model generalization results on fine-grained visual understanding benchmarks.

| Model | Dataset | Baseline | SiTu | Improvement |
|---|---|---|---|---|
| **InternVL 3.5 8B** | V* Bench | 64.0 | **72.0** | +8.0 |
| | HR-Bench 4K | 57.0 | **69.0** | +12.0 |
| | HR-Bench 8K | 54.0 | **65.0** | +11.0 |
| **Qwen2.5-VL 3B** | V* Bench | 69.0 | **76.0** | +7.0 |
| | HR-Bench 4K | 69.0 | **71.0** | +2.0 |
| | HR-Bench 8K | 66.0 | **72.0** | +6.0 |

HR-Bench are used for fine-grained perception, challenging models with high-resolution images (average resolution of $2246 \times 1582$ and 7680, respectively). We evaluate their sub-tasks—attribute recognition, spatial relationship reasoning, Fine-grained Single-instance Perception (FSP), and Fine-grained Cross-instance Perception (FCP)—using accuracy. For practical, real-world scenarios, we also use a subset of the MME-RealWorld benchmark, reporting on 12 representative sub-tasks.

**Implementation Details.** For our experiments, we use Qwen2.5VL-7B/3B and InternVL3.5 8B as the foundational Large Multimodal Model (LMM). All evaluations are performed on a single NVIDIA A40 GPU (48GB). To simplify and accelerate our experiments, each path uses at most one perception operation. As a baseline, we compare our framework against various closed- and open-source LMMs, training-based thinking with image methods, and other LMM perception enhancement methods.

## 3.2 RESULTS ON FINE-GRAINED VISUAL UNDERSTANDING

High-resolution benchmarks like $V^*$ Bench and HR-Bench present a significant challenge for VLMs due to their high image resolution and the small size of target objects. As shown in Table 1, our method, `SiTu`, achieves exceptional performance, surpassing both open-source models and complex, manually-engineered pipelines. On the $V^*$ Bench, `SiTu` achieves an overall accuracy of 92.1%, a significant improvement over the previous state-of-the-art. Our strong performance in attribute recognition (94.8%) highlights the framework's ability to effectively leverage fine-grained visual details. For the even more challenging HR-Bench, `SiTu` achieves an impressive overall accuracy of 79.5% on the 4K subset and 75.0% on the 8K subset. These results are notably higher than those of other methods. Specifically, on the Fine-grained Single-instance Perception (FSP) task, SiTu scores 95.0% and 92.0% on the 4K and 8K subsets, respectively. Compared to our foundational model, Qwen2.5-VL-7B, `SiTu` demonstrates remarkable performance gains of +20.9% on $V^*$ Bench and +10.7% on HR-Bench 4K. These results underscore the effectiveness of our approach in enhancing perception for high-resolution images, without the need for complex pipeline design or extensive training.

## 3.3 RESULTS ON HIGH-RESOLUTION PRACTICAL SCENARIOS

As shown in Table 2, our proposed `SiTu` method significantly boosts the performance of the baseline Qwen-2.5-VL-7B model on the MME-RealWorld benchmark. We see the largest gains in the MO/Calculate (+10.4%), MO/Color (+12.2%), and MO/Location (+18.8%) sub-tasks, highlighting `SiTu`'s effectiveness in improving a model's ability to perceive fine-grained details in complex, high-resolution scenes. However, the model's performance improvements were more limited in other areas, with smaller gains in the AD and OCR categories, such as AD/Visual Relation (+2.2%) and OCR/Text (+1.4%). We believe these results point to the inherent complexity of these tasks; for AD/Visual Relation, the challenge lies not just in perceiving multiple objects but in understanding their spatial relationships, while OCR/Text performance is often limited by the quality of the visual data itself. These findings show that while `SiTu` is highly effective at enhancing a model's high-resolution perception, there are still challenges in tasks that require complex reasoning or robust handling of degraded visual data.

Table 4: Action Space Extensibility and robustness analysis on V* Bench subset.

| Strategy | Overall Accuracy | Δ vs. Baseline | Role |
|---|---|---|---|
| Direct Answer (Naive Baseline) | 71.7% | - | - |
| + Contrast Enhancement | 70.2% | -1.5% | *Noisy/Harmful* |
| + Quadrant Selection | 74.3% | +2.6% | Beneficial |
| + Object Shrink | 76.4% | +4.7% | Beneficial |
| **SiTu (Combined Policy)** | **79.6%** | **+7.9%** | **Robust Integration** |

### 3.4 GENERALIZATION AND EXTENSIBILITY ANALYSIS

To demonstrate the universality and robustness of our framework beyond the primary setting, we conducted extensive additional experiments focusing on model generalization and action space extensibility.

**Generalization Across Architectures and Scales.** We validated SiTu on two additional models: InternVL 3.5 8B, which represents a distinct architectural design from the Qwen family, and Qwen2.5-VL 3B, representing a smaller parameter scale with weaker base reasoning capabilities. As shown in Table 3, SiTu consistently unlocks significant performance gains across all settings. Notably, on the challenging HR-Bench 4K, SiTu boosted InternVL's accuracy by **+12.0%**, proving that our uncertainty-guided perception enhancement is model-agnostic and effective across different architectures.

**Action Space Extensibility and Robustness.** To address concerns regarding the predefined action space, we prompted a frontier model (Gemini 3 Pro) to automatically generate three new perception strategies: **Quadrant Selection** (5-region zoom), **Object Shrink** (focused cropping), and **Contrast Enhancement** (global transformation). We integrated these unseen actions into SiTu and evaluated them on a subset of V* Bench.

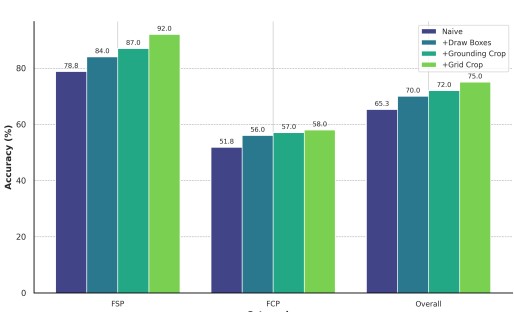

Figure 3: Comparison of method accuracy with respect to the addition of perception actions. As the number and diversity of actions increase, the method's performance exhibits stable gains, which demonstrates a scaling phenomenon for fine-grained visual understanding tasks on test-time. The x-axis represents different classes of subtasks, while the y-axis represents the accuracy percentage.

The results in Table 4 highlight two key findings: (1) **Synergy:** The combined SiTu policy (79.6%) outperforms the best single strategy (76.4%). (2) **Robustness:** Although *Contrast Enhancement* was a "noisy" strategy that individually degraded performance (-1.5%), incorporating it did not harm the SiTu system. Our uncertainty-guided mechanism successfully filtered out the low-quality outputs from this path, demonstrating the framework's ability to safely scale with diverse and even imperfect tools.

### 3.5 ABLATION STUDIES

To understand the contribution of each component of our framework, we conduct a series of ablation studies focusing on two key aspects: the effectiveness of individual perception strategies and the impact of our uncertainty-guided selection mechanism.

**Perception Actions Scaling Phenomenon.** To examine how the performance of `SiTu` evolves as more perception actions are incorporated, we start from the naive baseline and incrementally add the predefined actions: Draw Boxes, Grounding Crop, and Grid Crop. As shown in Figure 3, we observe a consistent performance improvement with the increase of actions, for both FSP and FCP. This demonstrates that the framework can effectively leverage action diversity to boost performance, while the cost of implementing such predefined perception operations is significantly lower than constructing extensive annotated datasets for training. Notably, in our action space, Grounding Crop and Grid Crop are functionally similar, both serving as irrelevant information suppression. However, the inclusion of Grid Crop does not lead to diminishing performance gains which is common in

ensemble mechanism (Fu et al., 2025). We attribute this to the distinct implementation mechanisms of the two crop actions, and more importantly, it suggests that `SiTu` is still far from reaching its upper limit in terms of supported actions diversity.

**Individual Perception Actions.** To determine whether our method's performance comes from specific perception actions or our uncertainty-guided mechanism, we compared the full `SiTu` method against the LMM naive answer and individual perception actions followed by naive answer. As shown in Table 5, perception actions can sometimes improve the original answer but may also introduce drawbacks. For instance, while Grounding Crop and Grid Crop improve FSP, they substantially reduce FCP, likely due to the crop operation tendency to discard visual information. Moreover, we observe a clear performance gap between the naive and single-action baselines and the full `SiTu`. This shows that the improvements of `SiTu` are not

Table 5: Ablation study of individual perception actions on HR-Bench 8K. Performance is measured by FSP, FCP, and Overall Accuracy. Full model performance is in bold.

| Method | FSP | FCP | Overall |
|---|---|---|---|
| Naive Answer | 78.8 | 51.8 | 65.3 |
|   - Draw Boxes | 80.0 | 52.0 | 66.0 |
|   - Grounding Crop | 83.0 | 47.0 | 65.0 |
|   - Grid Crop | 84.0 | 49.0 | 66.5 |
| `SiTu` (Full Method) | **92.0** | **58.0** | **75.0** |

attributable to single perception action. Instead, it combines multiple strategies to achieve superior overall performance, validating the effectiveness of the proposed uncertainty-guided architecture.

**Uncertainty-guided Metric.** To evaluate the effectiveness of our uncertainty-guided selection metric, we compare our framework's performance with five alternative metric methods: random selection, majority voting, perplexity, min entropy, and max entropy. As shown in Table 6, all candidate metrics within the uncertainty-guided framework achieve a performance improvement over the Naive Answer baseline, which demonstrates the stability of our uncertainty-guided framework. Furthermore, the average entropy-based approach yields the best results, showcasing its superiority over other metrics. We believe this advantage stems from the inherent definition of entropy and the averaging operation's comprehensive consideration of the influence of different tokens. Notably, this advantage even surpasses perplexity, which is typically used as an optimization objective during training.

Table 6: Comparison of different uncertainty metric mechanisms on HR-Bench 8K. Performance is measured by FSP, FCP, and Overall Accuracy. Our method is in bold.

| Selection Method | FSP | FCP | Overall |
|---|---|---|---|
| Random Selection | 83.0 | 49.0 | 66.0 |
| Majority Voting | 91.0 | 51.0 | 71.0 |
| Perplexity | 92.0 | 55.0 | 73.5 |
| Min Entropy | 85.0 | 53.0 | 69.0 |
| Max Entropy | 91.0 | 53.0 | 72.0 |
| Mean Entropy (`SiTu`) | **92.0** | **58.0** | **75.0** |

### 3.5.1 CASE STUDIES

To provide a more intuitive understanding of our framework's perception actions, we visualize several representative cases in Figure 4. The first row shows a typical example of our Draw Boxes action. When a query involves objects with significant size differences, a cropping-based approach often leads to information loss. Draw Boxes, on the other hand, highlights key areas while preserving the surrounding visual context, helping the LMM reduce hallucinations. The second row demonstrates the effectiveness of Grounding Crop. This action efficiently removes irrelevant content by using grounding to identify and crop crucial visual information that may be scattered across different locations in a high-resolution image, thus avoiding the information loss associated with fixed cropping methods. The third row illustrates the Grid Crop action. While less flexible than Grounding Crop, it retains more background information and offers different perspectives. This can be complementary to the other methods by providing additional context. Collectively, these cases demonstrate that each of our framework's perception actions possesses unique advantages. Compared to a naive approach, these actions accurately seek out crucial visual information, enabling the LMM to focus and respond to queries with enhanced precision.

## 4 RELATED WORKS

**Large Multimodal Models (LMMs).** Significant strides have been made in the field of Large Multimodal Models (LMMs), demonstrating substantial proficiency in a wide array of vision-language tasks (Bai et al., 2025a; Chen et al., 2024a; Liu et al., 2024a; Li et al., 2024; Team et al., 2025). The

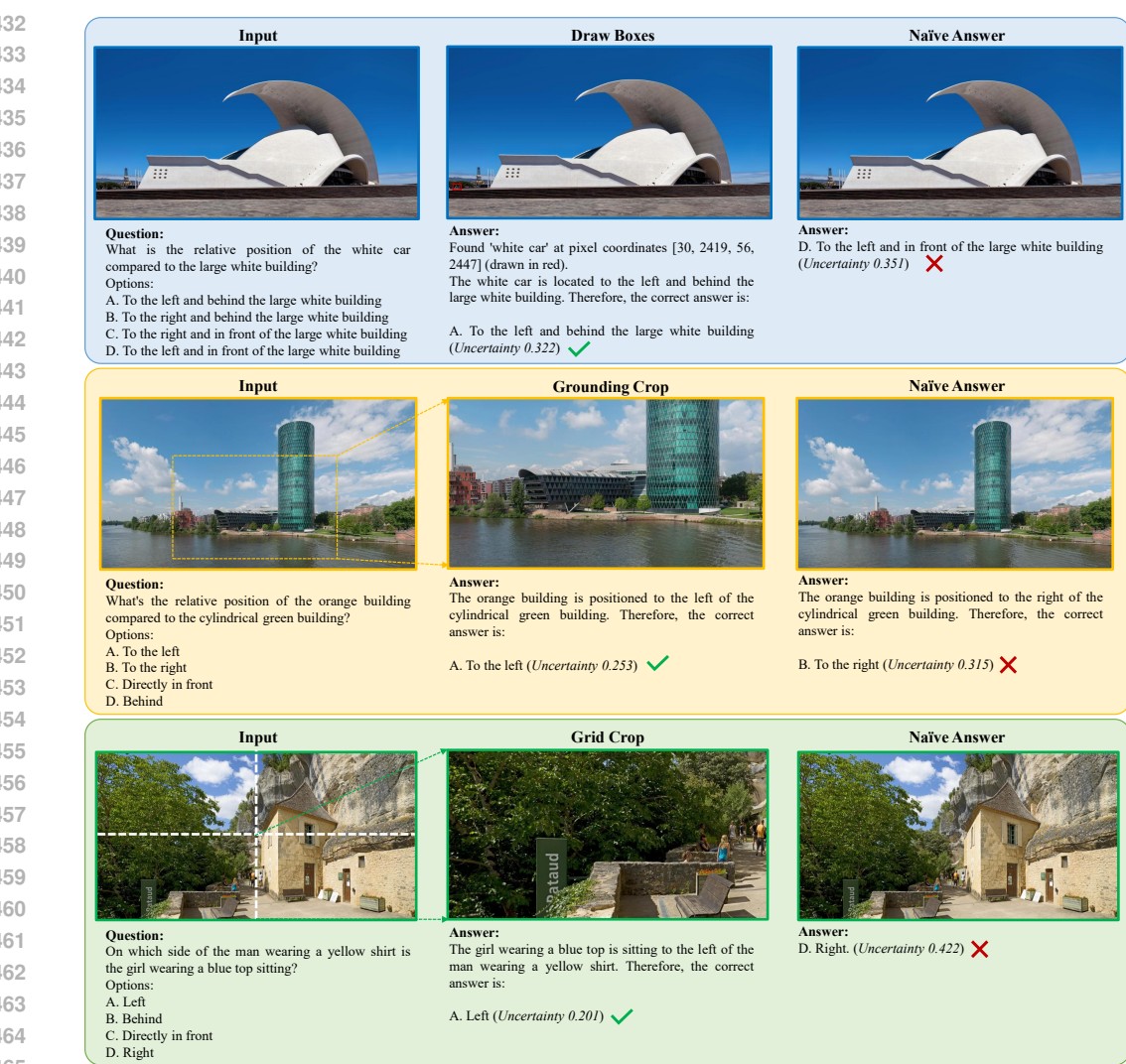

Figure 4: Visualization of Perception Actions. This case study is from the fine-grained visual understanding benchmark HR-Bench 8K. Each sample includes the original image and question, the intermediate image results from perception actions, and the final answer. A comparison with the naive answer effectively highlights the distinct advantages of each perception operation.

rapid evolution of this domain is underscored by the emergence of powerful open-source models, such as LLaVA (Liu et al., 2024a), InternVL (Chen et al., 2024c), and Qwen-VL (Bai et al., 2023a), which have achieved performance levels comparable to their closed-source counterparts. These models typically operate by fusing visual representations from specialized encoders with linguistic tokens, thereby enabling them to process and comprehend information across modalities (Liu et al., 2024c; Li et al., 2023). This capability has been pivotal in bridging the cognitive divide between visual perception and linguistic abstraction, allowing LMMs to perform sophisticated reasoning on a variety of multimodal challenges. Our work provides an orthogonal solution by demonstrating that MLLMs can be guided to actively explore and manipulate visual information in a training-free manner, thereby improving their perception and reasoning capabilities.

**Thinking with Images.** A new paradigm in multimodal reasoning moves beyond static inputs by enabling models to actively manipulate visual information. These approaches primarily rely on specialized training to instill dynamic perception capabilities directly into the model's weights. This is often achieved through supervised fine-tuning (SFT) (Wu & Xie, 2024; Wang et al., 2025c; Zhan et al., 2025) or reinforcement learning (RL) (Zheng et al., 2025b; Su et al., 2025a; Zhang et al.,

2025c;d). While these methods have shown promising results, they are fundamentally constrained by their high training costs and the limited generality of their learned operations. In stark contrast, our framework requires no training and is compatible with a wide range of perception operations, providing a flexible alternative that trades inference time for enhanced precision.

**Training-free Test-time Scaling.** Training-free test-time scaling methods enhance Large Language Model (LLM) reasoning at inference without requiring additional training. A prominent approach is self-consistency or parallel thinking, where multiple reasoning paths are generated and their final answers are aggregated, typically through majority voting (Wang et al., 2022). While this significantly boosts accuracy, it incurs a substantial computational cost, as generating numerous traces scales inference overhead linearly (Xue et al., 2023). However, this approach has limitations; its performance often plateaus or degrades as the number of traces increases. The core issue is that standard majority voting treats all traces equally, failing to account for quality variations. It is crucial to distinguish SiTu from prior uncertainty-based works in LLMs, such as ARPO (Dong et al., 2025) or FR3E (Zheng et al., 2025a). These methods primarily leverage high entropy to drive exploration during RL training to optimize textual reasoning policies. In contrast, SiTu employs entropy minimization as an exploitation signal during test-time inference, specifically targeting the multimodal perception bottleneck (i.e., visual grounding errors) rather than logical reasoning paths.

**Training-free LMM Perception Enhancement** Training-free methods for LMM perception enhancement leverage a model's existing capabilities to improve its processing of visual information without the need for additional fine-tuning or architectural changes. These approaches often employ agent-like workflows to guide the model's reasoning. For instance, methods like Dyfo (Li et al., 2025) and Zoom Eye (Shen et al., 2024) are inspired by human cognitive processes like visual search and dynamic zooming. Dyfo uses a Monte Carlo Tree Search to guide the model's focus to key visual regions, while Zoom Eye treats an image as a hierarchical tree to enable "vision-level reasoning." Other methods, such as Retrieval-Augmented Perception (RAP) (Wang et al., 2025b), adapt techniques from large language models, like Retrieval-Augmented Generation (RAG), to retrieve and fuse relevant image crops for better high-resolution perception.

## 5 CONCLUSION

In this work, we present `SiTu`, a novel training-free paradigm for "thinking with images" that overcomes the limitations of current training-based approaches. While existing methods rely on costly fine-tuning, which can cause catastrophic forgetting and narrow a model's capabilities, our approach leverages an LMM's inherent, untapped potential at test time. We address a core challenge for LMM scaling: enhancing perceptual ability for fine-grained visual understanding. By discovering a universal, LMM-native uncertainty metric, we dynamically guide the model through optimal perception paths, integrating diverse actions from visual highlighting to irrelevant information suppression. On several fine-grained benchmarks, `SiTu` not only outperforms existing training-free methods but also surpasses all open-source training-based "thinking with images" approaches, proving that enhanced performance can be unlocked without expensive fine-tuning.

**Limitations and Future Work.** While our proposed framework, `SiTu`, demonstrates significant potential, we acknowledge several limitations that also present exciting avenues for future research. First, the current action space of `SiTu` is not comprehensive. Its success has only been validated on fine-grained visual understanding tasks. It remains an open question whether this uncertainty-guided approach can be generalized to a broader range of multimodal tasks. This will require designing appropriate and effective actions tailored to different problem domains. Second, the perceptual actions in this work were manually designed. This approach, while effective, inherently limits the full potential of the LMM. Future work could explore automated methods to discover and optimize suitable perception actions for specific tasks, which could lead to further performance gains and reduce the computational cost associated with action execution. Finally, our `SiTu` framework is primarily a zero-shot, training-free test-time scaling method. An interesting direction would be to investigate whether providing a small number of in-context examples, similar to few-shot learning, could be used to obtain even greater performance returns. This would explore a hybrid approach that leverages the best of both training-free and data-driven methods.

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

## A    REPRODUCIBILITY DETAILS

### A.1    PROMPTS

We use standardized prompts to enforce consistent JSON outputs for downstream processing. For **Visual Highlighting** and **Grounding Crop**, we utilize the prompt: `"{question}.\nPlease locate the relevant item(s) in the image with its bbox coordinates and its name and output in JSON format."`. For **Grid Verification**, the prompt is: `"Is there a {object_name} in this image?  Answer Yes or No."`. For the **Quadrant Selection** (extended action), the prompt is: `"The image is divided into 5 regions (1:TL, 2:TR, 3:BL, 4:BR, 5:Center).  Which single region is MOST likely to contain the visual information needed?  Reply with the number."`

### A.2    HYPERPARAMETERS

For the **Grid Strategy**, we employ a dynamic grid based on the image aspect ratio; for standard 4:3 images, we default to a **2x2 grid** (4 quadrants). Regarding the **Crop Strategy** (specifically for Grounding Crop), we calculate the union bounding box of all detected objects and crop a region equivalent to **1/4 of the original image area**, centered on this union box. This effectively provides a **2x resolution zoom** on the region of interest. For **Uncertainty Calculation**, the entropy for each token is computed using the distribution of the **top-5 log-probabilities**. Finally, to ensure deterministic candidate generation, we iterate through fixed **Seeds** [0, 1, 2, 3, 4] for each visual view.

### A.3    EXTENDED ACTION SPACE DETAILS

To validate the framework's extensibility, we integrated three additional actions generated by Gemini 3 Pro. First, **Quadrant Selection** implements a coarse-to-fine zoom by dividing the image into 5 logical regions (Top-Left, Top-Right, Bottom-Left, Bottom-Right, and Center), where the model selects the most relevant region for cropping and upscaling. Second, **Object Shrink** performs a focused crop strictly around the detected object's bounding box with minimal padding, effectively removing background context to isolate the target. Third, **Contrast Enhancement** applies a global visual transformation using PIL `ImageEnhance.Contrast` with a factor of 1.5, aiming to improve visibility in low-light or low-contrast scenarios without altering spatial geometry.

### A.4    FAILURE ANALYSIS

We categorize failure cases into two primary types. **Type I (Logical Deficit)** occurs when the model accurately perceives the visual content but fails due to a lack of external knowledge or complex logical reasoning capabilities (e.g., mathematical derivation). **Type II (Action Misalignment)** happens in rare cases where aggressive cropping (e.g., Object Shrink) removes background context required for global reasoning (e.g., determining time of day from shadows). However, our uncertainty mechanism effectively minimizes the selection of such misaligned paths.

## B    DECLARATION OF LLM

The content and initial draft of this paper were manually authored. We employed Gemini for text polishing and for minor formatting adjustments to some LaTeX tables.

