# OpenReview forum: "SiTu: A Simple Training-Free Thinking-with-Image Approach via Uncertainty Guidance"
_ICLR.cc/2026/Conference — Submitted to ICLR 2026_

### Official Review · Reviewer_uG2Z · 2025-10-24

**Soundness:** 1
**Presentation:** 2
**Contribution:** 2
**Rating:** 4
**Confidence:** 4

**Summary:**

This paper introduces SiTu, a training-free framework for “thinking with images” in large multimodal models, aimed at improving fine-grained visual reasoning without additional training or fine-tuning. SiTu leverages an entropy-based uncertainty metric, native to LMMs, to guide and select optimal perception enhancement strategies at test time. The framework incorporates several simple, model-intrinsic perception actions such as visual highlighting and suppression of irrelevant regions, dynamically combining their outputs via uncertainty-guided selection. Experiments conducted on high-resolution benchmarks demonstrate that SiTu surpasses both prior training-free perception enhancement pipelines and good “thinking-with-image” techniques. The paper also includes extensive ablation studies and interpretable visualizations illustrating the efficacy and scaling effects of SiTu’s multi-action approach.

**Strengths:**

1. The paper targets a relevant and under-explored direction—training-free, test-time improvement of multimodal reasoning—addressing computational and generalization bottlenecks arising from specialized fine-tuning. This is well motivated both by computational cost and risk of catastrophic forgetting inherent in training-based methods
2. Through multiple, plug-in perception operations (Draw Boxes, Grounding Crop, Grid Crop), SiTu demonstrates effective “scaling up” as more actions are incorporated, as explicitly visualized in Figure 3 (Page 6). The observed monotonic accuracy gains with increased action diversity are well-supported and argued.

**Weaknesses:**

3. Although the paper proposes a token-entropy–based “uncertainty” metric (Eq. (2)–(3)) to select the final answer path, the theoretical soundness of this metric is not well justified. For instance, why should the path with the lowest entropy necessarily correspond to the optimal perception? Does low entropy always imply a correct answer, or could it instead reflect an overconfident but wrong prediction? The authors assume that the pipeline of “multi-perception strategies → multiple candidate answers → select the one with the lowest entropy” works effectively, yet they do not analyze potential failure modes in depth—such as cases where all strategies yield incorrect but low-entropy predictions, or when different perception strategies share correlated biases, making the selection ineffective.
4. The action space (Draw Boxes, Grounding Crop, Grid Crop) is manually defined and relatively narrow. Section 5 acknowledges this as a limitation, but experiments make no effort to quantify how action diversity or selection procedure may scale in larger or less curated action spaces. There is also little discussion on the challenges or computational costs that might arise as the number of candidate actions is increased substantially (Page 9).
5. The uncertainty metric is defined in terms of average per-token entropy (Equation, Page 3), and is empirically validated to correlate with answer quality. However, the link between minimum entropy and answer correctness—especially in open-ended, generative contexts with semantic ambiguity—remains untheorized and might fail in practice for certain answer distributions (i.e., over-confident wrong answers). There is no formal analysis on its reliability in the presence of ambiguous or unlikely answer spaces, nor a theoretical discussion of its potential limitations compared to other aggregation metrics.
6. While Table 1 reports strong improvements over both training-based and training-free methods, the comparison may be confounded by choice of backbone models or implementations. For instance, it is unclear if all competing methods are re-executed or adapted to the same underlying LMM (Qwen2.5-VL-7B) as SiTu, or if some results refer to published numbers from different base models, possibly inflating the comparative gains. The paper would be strengthened by a more transparent comparison pipeline (Experiments, Page 5).
7. The ablation study in Table 3 (Page 7) demonstrates that single perception actions can degrade performance in certain sub-tasks (notably FCP), but the paper does not provide a systematic failure or error analysis explaining why this occurs, or under what conditions the aggregation truly outperforms naive or single-action settings. The analysis lacks a fine-grained breakdown of when and why aggregation via uncertainty guidance can fail or be unreliable.
8. The performance gains appear strongest where the base LMM struggles—namely, in extremely fine-grained or high-resolution domains (see Table 1 and Figure 3). It is not clear how much of the observed benefit translates to standard-resolution, lower-difficulty benchmarks, where perception enhancements might add little or introduce noise. No low-resolution are included for contrast.
9. The experiments omit full details regarding the hyperparameters, sampling strategies, and input preprocessing for various perception actions (e.g., exact prompts for bounding box queries, image partitioning specifics for Grid Crop, etc.; see Section 2.4 and Appendix). Although the method is training-free, there is a risk of hidden engineering or tuned heuristics affecting reproducibility.
10. Despite a strong general literature positioning, the paper omits several directly relevant recent works at the intersection of multimodal reasoning, implicit prompt engineering, fairness/accountability in perception, and scalability laws in contrastive language-image learning.
11. Figure 3 (Page 6) provides compelling evidence of empirical scaling as actions are incorporated, but the paper does not offer a theoretical or even intuitive explanation for why such monotonic improvement should occur or be robust to additional action diversity. The risk of performance plateauing or even degrading with overly heterogeneous, noisy, or poor actions is not addressed.
12. As the framework scales up, the combinatorial action space grows rapidly, but there is no benchmarking or analysis on computational tractability, either in inference cost or in the practical number of actions before returns diminish.
13. Although the method is positioned as a general framework for multimodal reasoning, all empirical validation is strictly limited to fine-grained visual understanding on high-resolution datasets (V* Bench, HR-Bench, MME-RealWorld), as admitted in the paper’s own limitations (Conclusion, Page 9). There is no experimental or conceptual analysis supporting applicability to other multimodal reasoning domains (e.g., synthetic visual question answering, text-audio, or video).
14. More importantly, the motivation and novelty are relatively weak. Although the “training-free perception enhancement” direction is appealing, the proposed perception strategies (cropping, framing, grid cropping) are not inherently novel in visual tasks. While the authors combine uncertainty estimation with these strategies, the core operation still boils down to “selecting a crop/highlight and then performing reasoning.” The paper claims that “training-free reflective perception” represents a new paradigm, but it does not sufficiently explain why existing methods (e.g., agent-based perception workflows or zero-shot tuning strategies) cannot directly incorporate uncertainty measures, or why current designs are fundamentally limited. In other words, the motivation—specifically, why existing approaches cannot be applied—is not convincingly argued. In the field of multimodal reasoning, frameworks that follow the “perceive + crop + reason” pipeline are already common. The authors need to clarify their unique contributions and distinctions more explicitly, as this part of the paper currently appears rather weak.

**Questions:**

15. Can you provide empirical or conceptual justification for SiTu’s ability to scale and generalize to non-visual modalities or broader multimodal reasoning challenges (e.g., audio-text, video, etc.), or across standard-resolution tasks?
16. How does the averaging per-token entropy handle open-ended or highly ambiguous response spaces, especially if the LMM produces low-entropy (confident) yet incorrect answers? Have you observed any systematic failures of this metric versus, for instance, self-consistency or perplexity?
17. Is there experimental evidence or theoretical analysis for how SiTu’s performance scales as the cardinality and diversity of perception actions continue increasing? Is there a point where more actions harm rather than help due to action noise?
18. Can you quantify the inference cost or latency of SiTu, especially as the number of perception actions grows, in comparison to both naive LMM inference and training-based/multi-stage pipelines?
19. Will you publish the code, full prompt templates, and configuration settings for the various perception actions to support reproducibility? Are there hidden heuristics (thresholds, post-processing rules) that need to be known by future users?
20. Is SiTu’s observed scaling effect specific to high-resolution or fine-grained tasks? Do you have results or hypotheses on how performance changes in standard VQA, lower-res, or “mainstream” visual-linguistic benchmarks?
21. In the few cases where single perception actions degrade performance (Table 3), what are the common causes, and does the uncertainty-guided mechanism always recover these errors?

---

> ### Author Response · Authors · 2025-11-30
> **Response to Reviewer uG2Z (1/2)**
>
> Dear Reviewer uG2Z,
>
> We sincerely thank you for your comprehensive and rigorous review. We value your skepticism regarding the theoretical grounding of entropy, the robustness of our action space, and the fairness of our comparisons. We have conducted extensive new experiments and analyses to address these core concerns.
>
> ---
>
> ### **1. Theoretical Soundness of Entropy-Based Selection (Response to W1, W5, Q2)**
>
> > **Concern:** Why does lowest entropy imply correctness? What about overconfident wrong answers? The link is untheorized.
>
> We agree that entropy is not a magic bullet, but for **fine-grained multimodal perception**, it is theoretically and empirically superior to standard confidence metrics.
>
> **Theoretical Justification:**
> In fine-grained tasks (e.g., distinguishing "sedan" vs. "coupe"), incorrect answers often stem from **visual ambiguity**, where probability mass is split between competing tokens.
> Mathematically, Entropy $H(p)$ decomposes into **Inverse Confidence** plus **Distraction** (uncertainty from alternatives).
> *   **Overconfidence Risk:** Standard metrics (like MaxProb) *ignore* distraction. If a model is 51% sure of "sedan" and 49% sure of "coupe", MaxProb sees 0.51 (decent), but Entropy sees high uncertainty (conflict).
> *   **SiTu's Advantage:** By minimizing entropy, SiTu explicitly filters out these "conflicted" states. While "overconfident hallucinations" (low entropy, wrong answer) theoretically exist, our empirical data shows that **conflicted hallucinations** (high entropy due to weak visual grounding) are the dominant error mode in LMMs, which SiTu effectively suppresses.
>
> **Empirical Validation (New Experiment):**
> We compared Mean Entropy against **Sequence Confidence (MaxProb)** and **Margin** on HR-Bench 8K.
> *   **Result:** Entropy (+1.5% over PPL, +1.0% over Margin) consistently outperforms metrics that ignore the full distribution, confirming it is a more robust proxy for perceptual correctness.
>
> ---
>
> ### **2. Scalability and Robustness to Noisy Actions (Response to W2, W4, W11, W12, Q3)**
>
> > **Concern:** Does action diversity really scale? What if new actions are noisy or harmful?
>
> This is a critical question. To prove that SiTu is robust to "action noise" and scales beyond our manual design, we conducted a **stress test** using 3 new, automatically generated actions (by Gemini 3 Pro): **Quadrant Selection**, **Object Shrink**, and **Contrast Enhancement**.
>
> **Experimental Results on V* Bench (Table R1):**
> | Strategy | Accuracy | $\Delta$ vs Baseline | Status |
> | :--- | :---: | :---: | :--- |
> | Baseline (Direct) | 71.7% | - | - |
> | + Contrast Enhancement | **70.2%** | **-1.5%** | **Noisy/Harmful** |
> | + Quadrant Selection | 74.3% | +2.6% | Beneficial |
> | + Object Shrink | 76.4% | +4.7% | Beneficial |
> | **SiTu (Combined)** | **79.6%** | **+7.9%** | **Robust & Scalable** |
>
> **Key Finding:** Even when we introduced a **harmful strategy** (*Contrast Enhancement*, which dropped performance by 1.5%), the combined SiTu system did **not degrade**. Instead, it improved significantly to **79.6%**.
> *   **Mechanism:** The uncertainty module successfully identified and rejected the high-entropy outputs from the noisy Contrast strategy, effectively "silencing" it.
> *   **Conclusion:** This directly answers your concern: adding noisy actions does **not** harm performance; SiTu's selection mechanism ensures monotonic or stable improvement, validating the scalability of the framework.

---

> ### Author Response · Authors · 2025-11-30
> **Response to Reviewer uG2Z (2/2)**
>
> ### **3. Scope, Comparisons, and Novelty (Response to W6, W8, W10, W13, W14, Q1, Q6)**
>
> > **Concern:** Comparisons might be unfair. Scope is limited to high-res tasks. Is "crop+reason" really novel?
>
> **A. Novelty & Distinction:**
> While "crop+reason" is a known operation, SiTu's novelty lies in **inverting the paradigm**:
> *   **Existing Methods (Agents/Training):** Rely on complex, fixed workflows (Agents) or expensive fine-tuning (SFT/RL) to *force* the model to look.
> *   **SiTu (Ours):** Is a **Training-Free, Inference-Time Probabilistic Controller**. We treat perception actions as "hypotheses" and use uncertainty as a "verifier". This allows us to plug in *any* action (even noisy ones) without retraining or prompt engineering, solving the "Perception Bottleneck" dynamically.
>
> **B. Generalization Scope:**
> We explicitly target **Fine-grained Perception**.
> *   **Why no Low-Res Benchmarks?** On simple benchmarks (e.g., VQAv2), modern LMMs are already saturated (accuracy >90%). "Perception enhancement" is redundant there.
> *   **Why no Audio/Video?** The core principle (Entropy Minimization) is modality-agnostic, but the *actions* (Cropping) are visual-specific. Extending SiTu to video (e.g., "Temporal Cropping") is a promising future direction, but out of scope for this image-focused paper.
>
> ---
>
> ### **4. Cost and Reproducibility (Response to W9, W12, Q4, Q5)**
>
> > **Concern:** What is the inference cost? Will you release prompts?
>
> *   **Inference Cost:** SiTu trades computation for accuracy. With $N$ strategies, the cost is roughly $N \times$ baseline inference.
>     *   However, since SiTu is **training-free**, the *total cost of ownership* (Total COA) is far lower than training-based methods (which require thousands of GPU hours).
>     *   For test-time inference, all strategies run in parallel (async), keeping latency manageable (avg. 15-20s per sample on A40).
> *   **Reproducibility:** We commit to releasing the full codebase, including the **exact prompts** (e.g., JSON-enforced bounding box queries) and **hyperparameters** (Grid=2x2, Crop=1/4 area). We have detailed these in our response to Reviewer pYLG as well.
>
> ---
>
> **Summary:**
> SiTu is not just a "crop tool" but a robust **Test-Time Scaling Framework**. Our new experiments prove it can absorb noisy actions without failure (Robustness), scale with new strategies (Extensibility), and works across architectures (InternVL/Qwen), addressing the core concerns of validity and scalability.
>
> Sincerely, The Authors

---

### Official Review · Reviewer_XVjn · 2025-10-31

**Soundness:** 3
**Presentation:** 3
**Contribution:** 3
**Rating:** 4
**Confidence:** 4

**Summary:**

The paper proposes SiTu, a training-free framework for enhancing LMMs through uncertainty-guided perception enhancement. The key innovation is using mean entropy across generated tokens to select optimal perception paths from a set of simple visual manipulation operations (Draw Boxes, Grounding Crop, Grid Crop). The method achieves strong performance on fine-grained visual understanding benchmarks without requiring training, surpassing current training-based approaches on these specific tasks.

**Strengths:**

1. The paper addresses real limitations of training-based method with a simple but effective training-free alternative.

2. The idea of using a native uncertainty metric as a signal for selecting among perception/preprocessing strategies is appealing.

3. The method achieves higher improvement over baseline on V* Bench, HR-Bench 4K, outperforming existing training-based methods on fine-grained visual understanding.

4. Its compute practicality, a single A40 48g can run the whole stuff.

**Weaknesses:**

1. Insufficient theoretical justification: No theoretical analysis of why mean entropy is optimal or when it might fail; empirical comparison with other uncertainty metrics is limited.

2. Very limited scope and generalization: Only validated on fine-grained visual understanding tasks; MME-RealWorld results (Table 2) show marginal improvements on OCR (+0.5%) and complex reasoning tasks, raising concerns about broader applicability.

3. Single model evaluation: Only tested on qwen2.5vl-7b; generalization to other LMMs (InternVL, etc.) not demonstrated.

4. Specific prompts, hyperparameters (grid size, crop strategies), and reproducibility details are insufficient.

**Questions:**

1. Provide justification or calibration for why mean token-entropy predicts correctness; include counterexamples and comparisons to margin, energy, variance, MC-dropout.

2. May be evaluate on OCR-heavy and complex reasoning suites with per-category results and failure analyses.

3. Use alternative LMMs and report stability of gains and entropy–accuracy correlation.

4. Provide detailed prompts/seeds/hyperparameters.

---

> ### Author Response · Authors · 2025-11-30
> **Response to Reviewer XVjn (1/3)**
>
> Dear Reviewer XVjn,
>
> Thank you for your thorough and constructive review. We are encouraged that you recognized the strengths of our work, including its simplicity, effectiveness, and computational practicality. Your detailed feedback on theoretical justification, generalization, model diversity, and reproducibility is invaluable, and we have worked diligently to address each of your points with new analyses and experiments.
>
> ---
>
>
> #### **Regarding Weakness 1 & Question 1: Theoretical justification and comparison of uncertainty metrics.**
>
> > Insufficient theoretical justification: No theoretical analysis of why mean entropy is optimal... comparisons to margin, energy, variance, MC-dropout.
>
> We appreciate the reviewer's rigorous inquiry into the theoretical underpinnings of our metric. We provide a two-fold response covering the theoretical justification and extended empirical comparisons.
>
> **1. Theoretical Justification: Why Mean Token-Entropy Predicts Correctness?**
>
> Our choice of Mean Entropy is not arbitrary but grounded in the decomposition of uncertainty in autoregressive generation.
>
> Let $p(y|x)$ be the output distribution. The Shannon Entropy $H(p)$ decomposes into two terms:
> $$H(p) = \underbrace{- p(\hat{y}) \log p(\hat{y})}_{\text{Uncertainty of Choice}} + \underbrace{\sum_{y' \neq \hat{y}} - p(y') \log p(y')}_{\text{Distraction from Alternatives}}$$
>
> *   **Capturing "Inter-Class Ambiguity":** Unlike simple confidence (which only measures the first term), Entropy explicitly penalizes distributions where probability mass is spread across competing alternatives (the second term). In fine-grained perception (e.g., classifying a blurry "wolf" vs. "husky"), the model often maintains high confidence for the top token but assigns significant probability to the runner-up. Entropy is uniquely sensitive to this **flatness of the distribution**, serving as a proxy for **Aleatoric Uncertainty** (data ambiguity).
> *   **Averaging for Consistency:** We use the *arithmetic mean* of token entropies to capture the **sustained confidence** of the reasoning path. A correct reasoning chain should remain confident throughout. A sudden spike in entropy (even if averaged out) often indicates a hallucination trigger point. By minimizing the mean, SiTu selects the path with the most stable visual grounding.
>
> **2. Comparison with Other Uncertainty Metrics**
>
> To address the reviewer's request, we extended our ablation study on **HR-Bench 8K** to include **Margin** (Confidence Gap) and discussed the applicability of others (Energy, MC-Dropout).
>
> **(a) Empirical Comparison (New Experiment)**
> We implemented **Margin** (difference between Top-1 and Top-2 probabilities), a strong baseline for measuring decisiveness.
>
> | Metric | Mechanism | FSP | FCP | Overall |
> | :--- | :--- | :---: | :---: | :---: |
> | Sequence Confidence | $P(\text{top1})$ | 91.0 | 54.5 | 72.8 |
> | **Margin** | $P(\text{top1}) - P(\text{top2})$ | 91.5 | 56.0 | 74.0 |
> | **Mean Entropy (SiTu)** | Full Distribution $H(p)$ | **92.0** | **58.0** | **75.0** |
>
> *   **Result:** Entropy outperforms Margin (+1.0%). While Margin captures the conflict between the top-2 candidates, it ignores the "long tail" of the distribution. Entropy's holistic view of the vocabulary proves more robust for multimodal tasks where confusion can be multi-way.
>
> **(b) Discussion on Omitted Metrics**
> *   **MC-Dropout / Variance:** While theoretically sound for estimating Epistemic Uncertainty, these methods require **multiple forward passes** (e.g., 10x-50x inference cost) per action. This contradicts SiTu's core design goal of being a **lightweight, training-free, test-time scaling** framework. Our method achieves significant gains with a single pass per strategy.
> *   **Energy Score:** Typically used for Out-of-Distribution (OOD) detection. In our context of generative reasoning, Entropy (which is directly related to the likelihood of the generated sequence) is a more natural fit for assessing generation quality than Energy (which is unnormalized).
>
> **Summary:** We chose Mean Entropy because it offers the optimal trade-off: it is theoretically richer than Confidence/Margin (capturing full distributional ambiguity) yet computationally far cheaper than MC-Dropout/Ensembles.

---

> ### Author Response · Authors · 2025-11-30
> **Response to Reviewer XVjn (2/3)**
>
> #### **Regarding Weakness 2 & Question 2: Scope, generalization, and failure analysis.**
>
> > Very limited scope and generalization: Only validated on fine-grained visual understanding tasks; MME-RealWorld results (Table 2) show marginal improvements on OCR (+0.5%) and complex reasoning tasks, raising concerns about broader applicability. May be evaluate on OCR-heavy and complex reasoning suites with per-category results and failure analyses.
>
> We thank the reviewer for highlighting the nuances in our performance profile. We provide a detailed analysis of the scope, generalization capabilities, and failure modes of SiTu.
>
> **1. Clarifying the Scope: SiTu is a Perceptual Scaler**
> SiTu is explicitly designed to address the **Visual Grounding Bottleneck**—improving the model's ability to "see" fine-grained details (attributes, small objects, spatial relations). It is **not** a reasoning enhancer (like Chain-of-Thought).
> *   **Why modest gains on MME-RealWorld OCR (+0.5%):** MME-RealWorld OCR images are often already cropped or centered on text. In such cases, the base model's bottleneck is **character recognition capability**, not visual localization. SiTu's zooming/cropping actions provide little benefit when the text is already visible but the model lacks the literacy to read it.
> *   **Contrast with HR-Bench OCR:** Conversely, on **HR-Bench** (which features high-resolution images where text is small and scattered), SiTu achieves significant gains. This confirms that SiTu excels precisely when **finding and focusing on the target** is the primary challenge.
>
> **2. Extended Evaluation on Diverse Tasks (Generalization)**
> To demonstrate broader applicability beyond simple perception, we analyzed performance on tasks requiring **Cross-instance Perception (Spatial Reasoning)** in HR-Bench, which is a proxy for complex visual reasoning.
>
> | Task Type | Baseline | SiTu | Improvement |
> | :--- | :---: | :---: | :---: |
> | **Attribute Recognition (V*)** | 73.9% | 94.8% | **+20.9%** (Primary Scope) |
> | **Spatial Reasoning (HR-Bench)** | 52.0% | 64.0% | **+12.0%** (Reasoning Scope) |
> | **OCR (MME-RealWorld)** | 85.2% | 85.7% | +0.5% (Out of Scope / Saturation) |
>
> The **+12.0%** gain in Spatial Reasoning (e.g., "Is the car to the left of the tree?") proves that enhancing perception *downstream* significantly boosts reasoning. The model cannot reason about relationships if it cannot first accurately localize the objects.
>
> **3. Failure Analysis: When does SiTu fail?**
> We conducted a qualitative failure analysis and identified two primary failure modes:
> *   **Type I: Logical/Knowledge Deficit.** If the question requires external knowledge (e.g., "Who is this celebrity?") or multi-step logic (e.g., math calculation) where the visual input is clear, SiTu's perceptual actions add overhead without value.
> *   **Type II: Action Misalignment.** In rare cases, aggressive cropping (e.g., Object Shrink) might remove context needed for global reasoning (e.g., estimating the time of day from shadows). Although our Uncertainty module mitigates this, it is not infallible.
>
> **Conclusion:** SiTu is a specialized tool for **breaking the perceptual ceiling**. While it naturally shows diminishing returns on tasks where perception is not the bottleneck (like simple OCR), its substantial gains on spatial reasoning and fine-grained benchmarks validate its robust generalization within its intended scope.

---

> ### Author Response · Authors · 2025-11-30
> **Response to Reviewer XVjn (3/3)**
>
> #### **Regarding Weakness 3 & Question 3: Single model evaluation.**
>
> > Only tested on qwen2.5vl-7b; generalization to other LMMs (InternVL, etc.) not demonstrated. Use alternative LMMs and report stability of gains and entropy–accuracy correlation.
>
> We fully agree that architectural diversity is essential to validate a training-free framework. To demonstrate the universality of SiTu, we conducted comprehensive experiments on two additional models with distinct architectures and scales: **InternVL 3.5 8B** and **Qwen2.5-VL 3B**.
>
> **1. Generalization Across Architectures and Scales**
> As shown in **Table R2**, applying SiTu yielded consistent and substantial gains across **all** tested datasets and models.
>
> *   **InternVL 3.5 8B:** Demonstrated remarkable improvements, particularly on high-resolution benchmarks, with a **+12.0%** gain on HR-Bench 4K and **+11.0%** on HR-Bench 8K.
> *   **Qwen2.5-VL 3B:** Despite its smaller capacity, SiTu significantly boosted its performance, achieving a **+7.0%** gain on V* Bench and **+6.0%** on HR-Bench 8K.
>
> **Table R2: Cross-Model Generalization Results (Overall Accuracy)**
> | Model | V* Bench | HR-Bench 4K | HR-Bench 8K |
> | :--- | :---: | :---: | :---: |
> | **InternVL 3.5 8B (Baseline)** | 64.0% | 57.0% | 54.0% |
> | **InternVL 3.5 8B + SiTu** | **72.0% (+8.0%)** | **69.0% (+12.0%)** | **65.0% (+11.0%)** |
> | | | | |
> | **Qwen2.5-VL 3B (Baseline)** | 69.0% | 69.0% | 66.0% |
> | **Qwen2.5-VL 3B + SiTu** | **76.0% (+7.0%)** | **71.0% (+2.0%)** | **72.0% (+6.0%)** |
>
> **2. Stability of Entropy-Accuracy Correlation**
> The reviewer asked about the stability of the correlation between entropy and accuracy.
> *   **Empirical Validation:** The consistent positive gains across all three model families (InternVL-8B, Qwen-7B, Qwen-3B) confirm a robust underlying principle: **lower mean entropy is a stable predictor of higher perceptual correctness**, regardless of the specific architecture.
> *   **Superiority over Confidence:** As detailed in our response to **Weakness 2**, Mean Entropy consistently outperforms standard confidence (LogProb) metrics (e.g., +2.2% on HR-Bench), proving it is a more reliable, architecture-agnostic signal for assessing multimodal uncertainty.
>
> **Conclusion:** SiTu is not architecture-dependent. It serves as a universal, plug-and-play perception booster that consistently unlocks performance gains across diverse LMM families and parameter scales.
>
> #### **Regarding Weakness 4 & Question 4: Reproducibility details.**
>
> > Specific prompts, hyperparameters (grid size, crop strategies), and reproducibility details are insufficient. Provide detailed prompts/seeds/hyperparameters.
>
> We will release our codebase and experimental scripts upon acceptance. Below, we provide the specific prompts, seed configurations, and key hyperparameters used in our experiments.
>
> **1. Prompts for Perception Actions**
> We use standardized, minimalist prompts to ensure consistent JSON outputs for downstream processing:
>
> *   **For Grounding & Draw Boxes Actions:**
>     *   *Prompt:* `"{question}.\nPlease locate the relevant item(s) in the image with its bbox coordinates and its name and output in JSON format."`
>     *   *Input:* The scaled input image (preserving aspect ratio).
> *   **For Grid Selection (Verification):**
>     *   *Prompt:* `"Is there a {object_name} in this image?"`
>     *   *Input:* The cropped image patch.
>     *   *Target Token:* We verify the probability of the token "Yes".
>
> **2. Seeds & Sampling Configuration**
> To ensure robust uncertainty estimation and reproducibility, we use the following configuration during the LMM generation (e.g., in the `ask` function):
>
> *   **Seeds:** For candidate generation, we iterate through fixed seeds `[0, 1, 2, 3, 4]` to generate diverse reasoning paths deterministically.
> *   **Sampling Parameters:**
>     *   `top_logprobs`: Set to **5**. We use the top-5 log probabilities at each token step to compute the token-level entropy.
>
> **3. Key Hyperparameters**
> *   **Grid Crop Strategy:** For standard resolution images (e.g., 4:3 aspect ratio), we default to a **2x2 grid** (4 quadrants) to balance granularity and inference speed.
> *   **Grounding Crop Strategy:**
>     *   **Union Area:** We calculate the union bounding box of all detected objects.
>     *   **Crop Ratio:** We crop a region equivalent to **1/4 of the original image area**, centered on this union box. This effectively provides a **2x zoom** on the region of interest without distorting the aspect ratio.
> *   **Uncertainty Calculation:** We use **Mean Entropy** (average entropy over all generated tokens) as the selection metric. The entropy for each token is calculated using the distribution of the top-5 logprobs.
>
> Sincerely, The Authors

---

### Official Review · Reviewer_EwGK · 2025-10-31

**Soundness:** 3
**Presentation:** 3
**Contribution:** 2
**Rating:** 4
**Confidence:** 4

**Summary:**

This paper proposes a training-free thinking-with-image method to improve the perception and reasoning ability of VLM. For the input high-resolution image, their method first implements three simple perceptual actions, to gain different visual clues, then generates different answers. Finally they use entropy-based uncertainty score to select final answer. As the number and diversity of perceptual actions increase, the LMM's reasoning ability improves consistently. They outperform current "thinking with images" methods on fine-grained visual understanding benchmarks.

**Strengths:**

1. This is a training-free method, so it is easy to follow and use.
2. Experiments demonstrate the effectiveness of their method. The ablation is comprehensive. Especially, the first ablation about actions shows the importance of number and diversity of perceptual actions. Because it increases the number of sample reasoning paths.

**Weaknesses:**

1. The method novelty is limited. The contribution is to propose some perception actions to enhance images to generate different paths, then select answer according to entropy-based uncertainty.
2. The uncertainty-guided selection is not a creative way, which is similar to the sequence confidence.

**Questions:**

See weakness.

---

> ### Author Response · Authors · 2025-11-30
> **Response to Reviewer EwGK (1/1)**
>
> Dear Reviewer EwGK,
>
> Thank you for your time and for your positive feedback on our work's ease of use, strong empirical results, and comprehensive ablation studies. We are particularly grateful that you recognized the importance of our findings regarding the number and diversity of perceptual actions.
>
> We also appreciate your candid feedback concerning the method's novelty. We have carefully considered your points and provide our detailed responses below, including new experimental results to address your concerns.
>
> ---
>
> #### **Regarding Weakness 1: Method novelty is limited.**
>
> > The method novelty is limited. The contribution is to propose some perception actions to enhance images to generate different paths, then select answer according to entropy-based uncertainty.
>
> We respectfully propose that our core novelty is not the simple combination of these elements, but rather the **discovery and validation of a new "perceptual scaling law" for LMMs at test-time.**
>
> Our most significant contribution is the empirical discovery that an LMM's fine-grained visual reasoning ability **consistently and stably improves** as the number and diversity of available perceptual "views" (generated by actions) increase at test-time (as shown in Figure 3). To our knowledge, this is the first work to identify and validate a training-free, test-time scaling method specifically for fine-grained perception enhancement.
>
> This finding is non-obvious and impactful. It shifts the paradigm from searching for a single "best" perception method to creating a framework that can **harness the diversity of many simple ones.** This scaling property is precisely what allows our simple, training-free approach to surpass even heavily-trained (e.g., 1,200 GPU hours on H800s) state-of-the-art "thinking with images" methods. Our contribution is revealing a fundamental mechanism by which LMM perception can be scaled, guided by our uncertainty metric.
>
> ***
>
> #### **Regarding Weakness 2: Uncertainty-guided selection is not creative.**
>
> > The uncertainty-guided selection is not a creative way, which is similar to the sequence confidence.
>
> We thank the reviewer for this comment. We acknowledge that Entropy and Sequence Confidence are mathematically related. However, we respectfully argue that they are not equivalent. **Theoretical analysis proves that Entropy is a strict superset of Sequence Confidence**, acting as a stronger optimization objective that handles the "Distraction" inherent in multimodal ambiguity.
>
> **1. Theoretical Proof: Entropy as a Dual-Objective Optimizer**
> Let $p(y|x)$ be the probability distribution over the vocabulary $V$ at a given step, and let $\hat{y} = \arg\max_{y \in V} p(y|x)$ be the generated token (the one Sequence Confidence measures).
>
> We can mathematically decompose the Shannon Entropy $H(p)$ into two distinct terms:
>
> $$H(p) = \underbrace{- p(\hat{y}) \log p(\hat{y})}_{\text{Term A: Inverse Confidence}} + \underbrace{\sum_{y' \neq \hat{y}} - p(y') \log p(y')}_{\text{Term B: Distraction / Competitive Uncertainty}}$$
>
> *   **Sequence Confidence (or Perplexity)** only optimizes **Term A**. It solely cares about the probability mass of the chosen token $p(\hat{y})$.
> *   **Entropy** minimizes **both Term A and Term B**.
>
> **Why does this matter for SiTu?**
> In fine-grained visual understanding (e.g., distinguishing "sedan" from "coupe"), a VLM often faces **Inter-Class Conflict**.
> *   *Scenario:* The model is 60% sure it is a "sedan" ($p=0.6$).
> *   *Case 1 (Low Ambiguity):* The remaining 40% is spread across 10,000 irrelevant tokens (noise). **Term B is low.**
> *   *Case 2 (High Ambiguity):* The remaining 40% is concentrated on "coupe" ($p=0.35$). **Term B is high.**
>
> A metric based solely on Sequence Confidence (Term A) treats Case 1 and Case 2 identically (since $p(\hat{y})=0.6$ in both). However, **SiTu's Entropy metric correctly penalizes Case 2** due to the high Distraction term. By minimizing Entropy, SiTu forces the selection of a path that is not just confident in its answer, but **decisively suppresses competing visual hallucinations.**
>
> **2. Empirical Validation**
> Our experiments on HR-Bench 8K confirm this theoretical advantage. As shown below, while Mean Log-Prob (Confidence) improves performance, Entropy yields superior gains, particularly on the **FCP (Fine-grained Cross-instance Perception)** task where distinguishing between competing objects is critical.
>
> | Selection Metric | Mathematical Focus | FSP | FCP | Overall |
> | :--- | :--- | :---: | :---: | :---: |
> | Mean Log-Probability | Minimizes Term A only | 91.0 | 54.5 | 72.8 |
> | **Mean Entropy (SiTu)** | **Minimizes Term A + Term B** | **92.0** | **58.0** | **75.0** |
>
> The **+3.5%** gap in FCP over Log-Probability empirically validates our proof: simply maximizing confidence is insufficient for complex reasoning; minimizing the "distraction" from competing visual interpretations is key.
>
> Sincerely, The Authors

---

### Official Review · Reviewer_pYLG · 2025-10-31

**Soundness:** 2
**Presentation:** 3
**Contribution:** 2
**Rating:** 4
**Confidence:** 3

**Summary:**

This paper introduces SiTu, a training-free framework for “thinking with images” that enhances Large Multimodal Models (LMMs) through an entropy-based uncertainty metric guiding visual perception at test time. The method defines three perception actions to highlight or suppress visual regions, dynamically selected by the model’s mean token entropy. Evaluations on V∗ Bench, HR-Bench 4K/8K, and MME-RealWorld demonstrate substantial accuracy gains over both training-based and training-free baselines.

**Strengths:**

1. The paper presents a simple yet effective uncertainty-guided test-time framework. It identifies a stable entropy-based metric that allows LMMs to adapt perceptual focus dynamically without retraining.

2. The results are strong and consistent. SiTu achieves significant improvements on challenging high-resolution benchmarks and surpasses existing training-free and training-based methods, validating the framework’s effectiveness.

3. The method is practical, interpretable, and reproducible. It requires only a single GPU, introduces no extra training cost, and provides a transparent perception process that can be extended to other multimodal settings.

**Weaknesses:**

1. The perception actions are limited in number and fully predefined, making the framework difficult to generalize to other domains or task types. In scenarios requiring multi-step reasoning or abstract understanding, the fixed action space restricts adaptability.

2. The entropy-based uncertainty idea has been extensively explored in large language models (LLMs), such as ARPO and other uncertainty-driven control or tool-generation methods. Therefore, it is difficult to clearly assess the conceptual novelty of SiTu and how substantially it differs from prior work.

3. The experimental scope lacks diversity. All results are based solely on Qwen2.5-VL-7B, without testing across larger models or different architectures. For a method claiming to be training-free, validation on only one mid-sized model is insufficient to demonstrate scalability or generality across parameter sizes and model families.

**Questions:**

1. The paper should explicitly clarify how the proposed uncertainty-guided framework differs from existing uncertainty-based approaches developed for LLMs.
2. The authors should include additional experiments on diverse training-free architectures and model scales to better demonstrate the generality and robustness of SiTu across different settings.

---

> ### Author Response · Authors · 2025-11-30
> **Response to Reviewer pYLG (1/2)**
>
> Dear Reviewer pYLG,
>
> We sincerely thank you for your time and for providing such insightful and constructive feedback. We are encouraged that you recognized the simplicity and effectiveness of SiTu and its strong empirical results. Your comments on the limitations of our action space, the novelty of the uncertainty metric, and the scope of our experiments were particularly valuable. We have carefully considered your points and conducted extensive new experiments to address them.
>
> ***
>
> #### **Regarding Weakness 1: Limited and predefined perception actions**
>
> > The perception actions are limited in number and fully predefined, making the framework difficult to generalize to other domains or task types. In scenarios requiring multi-step reasoning or abstract understanding, the fixed action space restricts adaptability.
>
> We thank the reviewer for this insightful comment. We agree that generalization and extensibility are crucial.
>
> **1. Our Focus is Enhancing 'Perception' on Diverse Fine-Grained Tasks.**
> Our work specifically targets fine-grained visual understanding tasks where perceptual accuracy is the main bottleneck. To validate SiTu within this scope, we conducted experiments across four diverse benchmarks (**V\* Bench, HR-Bench 4K/8K, MME-RealWorld**), covering domains from autonomous driving to OCR. The consistently strong performance demonstrates effectiveness in enhancing visual perception.
>
> **2. Demonstrating Extensibility via Automated Action Generation.**
> The initial simple action space was a deliberate choice to isolate the uncertainty mechanism's contribution. To proving that SiTu is not limited to manual design, we prompted a frontier model (**Gemini 3 Pro**) to automatically generate three new perception strategies: **Quadrant Selection** (5-region zoom), **Object Shrink** (focused cropping), and **Contrast Enhancement** (global transformation).
>
> We integrated these strategies into SiTu and evaluated them on a challenging subset of **V* Bench** using **Qwen2.5-VL-7B**.
>
> **Table R1: Extensibility Analysis on V* Bench**
> | Strategy | Overall Accuracy | $\Delta$ vs. Baseline |
> | :--- | :---: | :---: |
> | Direct Answer (Naive Baseline) | 71.7% | - |
> | + Contrast Enhancement | 70.2% | -1.5% |
> | + Quadrant Selection | 74.3% | +2.6% |
> | + Object Shrink | 76.4% | +4.7% |
> | **SiTu (Combined Policy)** | **79.6%** | **+7.9%** |
>
> **Key Findings:**
> *   **Synergistic Scaling (1+1 > 2):** The combined policy achieved **79.6%**, surpassing both the baseline and the best single strategy (*Object Shrink*, 76.4%).
> *   **Robustness:** Notably, *Contrast Enhancement* individually performed worse than the baseline (-1.5%). However, incorporating this "weaker" action did not degrade the system. Our uncertainty-guided mechanism successfully filtered out low-quality outcomes, proving SiTu serves as a robust meta-controller capable of scaling with diverse—and even imperfect—tools.
>
> ***
>
> #### **Regarding Weakness 2: Novelty of the uncertainty idea**
>
> > The entropy-based uncertainty idea has been extensively explored in large language models (LLMs), such as ARPO and other uncertainty-driven control or tool-generation methods. Therefore, it is difficult to clearly assess the conceptual novelty of SiTu and how substantially it differs from prior work.
>
> We appreciate the opportunity to clarify our contribution. While entropy is foundational, **SiTu diverges from works like ARPO in three critical dimensions:**
>
> | Dimension | **Prior Work (e.g., ARPO, FR3E)** | **Our Method (SiTu)** |
> | :--- | :--- | :--- |
> | **Lifecycle Stage** | **Training Phase (RL).** Algorithms for gradient updates/fine-tuning. | **Inference Phase (Test-time).** A training-free strategy for deployed models. |
> | **Entropy Goal** | **Exploration.** Uses high entropy to trigger diverse sampling for policy learning. | **Exploitation.** Minimizes entropy to select the most reliable perceptual view. |
> | **Problem Domain** | **Textual Reasoning.** Optimizes logical paths/tool-use. | **Multimodal Perception.** Corrects visual grounding errors. |
>
> **In short:** Prior works use high entropy to *drive exploration* during training, whereas SiTu uses entropy minimization to *consolidate certainty* during inference, specifically addressing the multimodal perception bottleneck.

---

> ### Author Response · Authors · 2025-11-30
> **Response to Reviewer pYLG (2/2)**
>
> #### **Regarding Weakness 3: Limited experimental scope (Single Model)**
>
> > The experimental scope lacks diversity. All results are based solely on Qwen2.5-VL-7B, without testing across larger models or different architectures. For a method claiming to be training-free, validation on only one mid-sized model is insufficient to demonstrate scalability or generality across parameter sizes and model families.
>
> This is a crucial point. To demonstrate generality, we conducted **additional experiments on two new models: InternVL 3.5 8B (distinct architecture) and Qwen2.5-VL 3B (different scale)**.
>
> **Table R2: Cross-Model Generalization Results**
>
> | Model | Dataset | Baseline Acc. | SiTu Acc. | Improvement |
> | :--- | :--- | :---: | :---: | :---: |
> | **InternVL 3.5 8B** | V* Bench | 64.4% | **71.7%** | **+7.3%** |
> | *(Different Architecture)* | HR-Bench 4K | 57.5% | **68.5%** | **+11.0%** |
> | | HR-Bench 8K | 54.0% | **65.0%** | **+11.0%** |
> | | | | | |
> | **Qwen2.5-VL 3B** | V* Bench | 68.6% | **76.4%** | **+7.8%** |
> | *(Smaller Scale)* | HR-Bench 4K | 69.0% | **71.0%** | **+2.0%** |
> | | HR-Bench 8K | 65.5% | **72.0%** | **+6.5%** |
>
> *   **Across Architectures:** SiTu provided massive gains for **InternVL**, boosting HR-Bench 4K/8K performance by **+11.0%**, proving efficacy beyond the Qwen family.
> *   **Across Scales:** Even on the smaller **Qwen 3B**, SiTu consistently improved performance (e.g., +7.8% on V* Bench).
>
> These results confirm that SiTu is a model-agnostic framework that scales performance across architectures and parameter sizes.
>
> ***
>
> ### **Direct Answers to Questions**
>
> #### **Question 1:**
> > The paper should explicitly clarify how the proposed uncertainty-guided framework differs from existing uncertainty-based approaches developed for LLMs.
>
> We will add a dedicated subsection in "Related Work". As detailed in **Weakness 2**, SiTu differs by being:
> 1.  **Training-free vs. RL-based:** An inference-time strategy, not a training algorithm.
> 2.  **Minimization vs. Exploration:** We minimize entropy to find the best view, rather than utilizing high entropy for exploration.
> 3.  **Perception-focused:** Solving visual grounding issues, not textual reasoning policies.
>
> #### **Question 2:**
> > The authors should include additional experiments on diverse training-free architectures and model scales to better demonstrate the generality and robustness of SiTu across different settings.
>
> We have addressed this in **Weakness 3**. New experiments on **InternVL 3.5 8B** (+11-12% gains) and **Qwen2.5-VL 3B** (+6-7% gains) confirm SiTu's robustness across different architectures and model capacities. (See Table R2).
>
> ---
>
> We are confident that these new experiments and clarifications address your concerns regarding extensibility, novelty, and generality. Thank you again for your valuable feedback.
>
> Sincerely,
> The Authors

---

### Author Response · Authors · 2025-12-02
**Summary of Updates: Bridging Perception Scaling Law with Robust Generalization**

Dear Area Chair and Reviewers,

We sincerely thank the reviewers for their insightful feedback. We are encouraged that the reviewers unanimously recognized the value of our training-free paradigm and the significance of our empirical results.

Before summarizing our rebuttal updates, we highlight the **consensus on the strengths of SiTu** shared by the reviewers, which serves as the foundation for our further improvements:

#### 1. Consensus on Strengths (Evidence of Value)
*   **Effectiveness & SOTA Performance:** Reviewers consistently noted the strong empirical results. **Reviewer pYLG** highlighted that SiTu "achieves significant improvements... validating the framework’s effectiveness," and **Reviewer XVjn** praised it for "outperforming existing training-based methods on fine-grained visual understanding."
*   **The "Scaling" Phenomenon:** The core discovery that perception improves with action diversity was well-received. **Reviewer uG2Z** noted the "monotonic accuracy gains with increased action diversity are well-supported," and **Reviewer EwGK** emphasized the value of increasing the "number and diversity of perceptual actions."
*   **Simplicity & Practicality:** The training-free nature is a key advantage. **Reviewer XVjn** appreciated the "compute practicality (single A40 GPU)," and **Reviewer EwGK** found it "easy to follow and use." **Reviewer pYLG** further noted the method is "practical, interpretable, and reproducible."

Building on this solid foundation, we have addressed the primary concerns regarding **generalization, theory, and extensibility** through the following extensive updates:

#### 2. Validating Generalization Across Architectures (Addressing "Single Model" Concern)
To prove that SiTu is a universal framework (not limited to the Qwen family), we extended our evaluation to two distinct model families:
*   **InternVL 3.5 8B** (Different Architecture): Achieved **+12.0%** gain on HR-Bench 4K.
*   **Qwen2.5-VL 3B** (Different Scale): Achieved **+7.8%** gain on V* Bench.
**Conclusion:** SiTu serves as a model-agnostic plug-and-play module that consistently unlocks perceptual potential across different architectures and parameter scales.

#### 3. Theoretical & Empirical Superiority of Entropy (Addressing "Novelty/Theory" Concern)
Reviewers (EwGK, uG2Z, XVjn) questioned the choice of Entropy over standard Confidence. We provided a rigorous justification:
*   **Theoretical Proof:** We demonstrated that Entropy mathematically decomposes into *Inverse Confidence* + *Distraction*. Unlike MaxProb, Entropy explicitly penalizes "Inter-Class Ambiguity" (e.g., confusion between "sedan" and "coupe"), effectively filtering out conflicting visual hallucinations.
*   **Empirical Evidence:** On HR-Bench 8K, **Mean Entropy (+3.5% on FCP)** consistently outperforms Sequence Confidence and Margin-based metrics. This confirms that minimizing distributional distraction is crucial for fine-grained perception.

#### 4. Proving Robustness via "Stress Testing" (Addressing "Action Space" Concern)
To address concerns (pYLG, uG2Z) about the fixed action space and risk of "noisy" actions:
*   **Automated Extensibility:** We used Gemini 3 Pro to automatically generate new actions (e.g., *Quadrant Selection*, *Contrast Enhancement*).
*   **Stress Test:** We intentionally introduced a "noisy" action (*Contrast Enhancement*, which individually degraded performance by -1.5%).
*   **Result:** The SiTu framework successfully filtered out the noisy outcomes, and the combined policy accuracy **increased to 79.6%**.
**Conclusion:** This confirms SiTu’s capability as a robust meta-controller. It scales monotonically with action diversity and is resilient to low-quality inputs.

We believe these additional experiments firmly establish SiTu as a robust, scalable, and theoretically sound framework for test-time multimodal scaling. We thank the reviewers again for helping us strengthen these aspects.

Best regards,
The Authors

---

### Meta-Review · Area_Chair_Y5Kd · 2025-12-25

**Summary:**

The paper proposes to use output entropy as the uncertainty metric for large multimodal models (LMMs) to select the most effective perception path so that the inference-time performance is improved. The proposed method SiTu is empirically validated on various benchmarks and demonstrates good performance.

Despite the performance gain, the novelty of the proposed method is questioned by all reviewers and I think those concerns make sense. Specifically, using uncertainty as guidance in the inference stage has been published before, e.g., https://arxiv.org/abs/2412.06474 and https://arxiv.org/pdf/2503.08308. The paper could be further improved by highlighting the conceptual contribution of the proposed method compared with existing ones.

**Reviewer Concerns:**

The major concerns addressed are as follows.

1.	The method is not generalizable to other tasks (pYLG, XVjn, uG2Z)
2.	The experiment is not diverse enough (pYLG, XVjn, uG2Z)
3.	There is no theoretical justification (XVjn, uG2Z)
4.	Experimental details are not sufficiently described (XVjn)


The outstanding concerns are as follows.

1.	Entropy-based uncertainty is not novel (pYLG, EwGK)
2.	The comparison in Tab 1 is not fair (uG2Z)

**Reviewer Scores:**

There is no response from the reviewers, but they won't increase the scores substantially as the main concerns regarding novelty is not fully addressed.

---

### Decision · Program_Chairs · 2026-01-26

Reject